# Analysis of Factors Influencing the Job Satisfaction of New Generation of Construction Workers in China: A Study Based on DEMATEL and ISM

**Guodong Ni [1],\***[ORCID]**, Huaikun Li [2], Tinghao Jin [2], Haibo Hu [3] and Ziyao Zhang [2]**

[1]  Research Center for Digitalized Construction and Knowledge Engineering, School of Mechanics and Civil Engineering, China University of Mining and Technology, Xuzhou 221116, China

[2]  School of Mechanics and Civil Engineering, China University of Mining and Technology, Xuzhou 221116, China; lihuaikun@cumt.edu.cn (H.L.); jintinghao@cumt.edu.cn (T.J.); zhangziyao@cumt.edu.cn (Z.Z.)

[3]  Department of Architectural Engineering, Xuhai College, China University of Mining and Technology, Xuzhou 221116, China; huhaibomolan@163.com

\*  Correspondence: niguodong@cumt.edu.cn

**Abstract:** China's construction industry is facing serious problems of aging construction workers and labor shortages. Improving the job satisfaction of construction workers is a key point for retaining existing construction workers and for attracting younger generations into the construction field in China. At present, the new generation of construction workers (NGCW) born after 1980 has been the main force on construction sites in China. Therefore, it is very important to study and explore the influencing factors of the job satisfaction of the NGCW. This paper aims to determine the influencing factors of job satisfaction of the NGCW through literature research and to clarify the interaction mechanisms and hierarchical structures of influencing factors using the Decision-Making Trial and Evaluation Laboratory (DEMATEL) and Interpretive Structural Modeling (ISM) to design appropriate human resource practices to promote their job satisfaction. Research findings show that there are 12 main influencing factors of job satisfaction of the NGCW, which are at three levels: personal traits, job characteristics and social environment, and the influencing factors can be divided into a cause group and an effect group, including four layers: the root layer, controllable layer, key layer and direct layer in the multi-level hierarchical structure model. Furthermore, the critical influencing factors of the job satisfaction of the NGCW consist of education level, competency, career development, salaries and rewards, rights protection and work–family balance. This research enriches the job satisfaction literature of construction workers and provides an important reference for decision makers in construction enterprises and the construction industry to understand what influences the job satisfaction of the NGCW and how it is influenced to then improve it in China.

**Keywords:** job satisfaction; influencing factor; new generation of construction workers; DEMATEL-ISM

## 1. Introduction

As a mainstay industry for the overall development of countries, the construction industry is well known for its labor intensity [1–4]. In the past few decades, there have been plenty of migrant laborers who have poured into the construction industry in China [5–7]. Currently, the aging trend of migrant workers is relatively significant, and the proportion of migrant workers over the age of 50 has risen to 26.4% in China [8]. However, the construction industry is physically demanding for workers [9,10]. Therefore, older generations of construction workers have been gradually leaving due to increases in age and declines in physical fitness [7,11,12]. Moreover, the administrative department of construction has formulated and issued policies in China, which require the dismissal of over-age migrant workers (over 60) on site [13]. As a result, the number of employees in China's construction industry has declined for three consecutive years up to 2021, with a decrease of 0.84 million

compared with the previous year [14]. In other words, the construction industry is facing a serious labor shortage, which is quickly becoming a global phenomenon [5,15–18], and this poses considerable threats to the construction industry [19–21], such as rising costs and delayed deliveries [22]. Aging and labor shortage problems reveal the unwillingness of younger generations to enter the construction industry even though they can play a strategic role in establishing and maintaining the labor force of the construction industry [7,16,23]. Therefore, it is very important to explore how to retain existing construction workers and how to attract younger generations to join the construction industry.

The existing literature shows that job satisfaction demonstrates a correlation with labor market behaviors, such as resignation, absenteeism, burnout and retention [24–29]. It is believed that job satisfaction plays an important role in attracting potential employees and gluing employees to their organization [30–32]. Above all, low job satisfaction may explain why young people are reluctant to engage in the construction industry. Hence, improving the job satisfaction of construction workers is a key point for retaining existing construction workers and for attracting younger generations into the construction field in China.

In recent years, the new generation of migrant workers born after 1980, as a special group growing up in an important period of China's reform, has attracted the attention of many researchers in China because it has become the main body of workers and has made outstanding contributions to social development [6,33–35]. Migrant workers are a major workforce in China's construction industry, who accounted for a large proportion (18.3%) of total migrant workers (285.6 million) in 2020 [8]. The new generation of construction workers (NGCW) who were born after 1980 and who engaged in the construction industry as laborers or skilled workers has been the main force on construction sites in China at present [36]. Therefore, it is necessary to study the job satisfaction of this specific group who were born and raised in a period of China's economic and educational reform and development.

Compared with the older generation, most of the NGCW has a rural household registration but does not earn a living from farming [6], and these workers also have typical characteristics of the new generation of migrant workers, such as better education [37], better learning capability [38], a greater emphasis on work experience [39], a stronger awareness of rights protection [40] and lower endurance for work [41]. Moreover, their personal needs have also changed with socio-economic growth and with improvements on quality of life, thus pursuing higher-order needs such as self-actualization [42], placing more importance on freedom [15], showing a stronger tendency towards individualism and consumerism [41], paying more attention to social status [5], etc. For example, they may consider accommodation, leisure and even training opportunities as critical factors in choosing jobs [5]. This may be related to growth in education level, which brings advancement in professional goals [43]. Therefore, differences in personal characteristics and changes in needs may make the psychological threshold of the job satisfaction of the NGCW higher than that of the older generation. Furthermore, the construction industry is often associated with a poor image, such as being dirty and dangerous, having low status, long working hours, bad work safety and health conditions and an ambiguous career path [5,23,27,44,45]. As a result, these discrepancies and contradictions between psychological expectations and actual conditions have led to the job satisfaction of the NGCW becoming the focus of human resource management in construction enterprises in China. In addition, identifying the influencing factors of the job satisfaction of the NGCW and analyzing their interaction mechanisms have become the key to solving this practical issue.

Existing studies mainly focus on the consequences of job satisfaction, such as knowledge sharing [46], safety behavior [36], safety climate perceptions [47] and perceived health [48]. Limited studies on influencing factors mainly consider situational factors, including job characteristics and job conditions [28]. In addition, researchers have made certain achievements on the influencing factors of the job satisfaction of employees in the construction industry, such as project managers [29,32,49], quantity surveyors [50] and

construction engineers [51]. However, less attention is paid to construction workers and their intergenerational differences. In addition, the applicability of research based on the job satisfaction of managers to workers needs testing [52]. In other words, few research efforts have been made regarding the influencing factors of the job satisfaction of the NGCW. Considering its unique group characteristics and needs, this paper aims to determine the influencing factors of job satisfaction of the NGCW and to clarify the interaction mechanisms and hierarchical structures of influencing factors for the purpose of designing appropriate human resource practices to promote job satisfaction. From the perspective of generational differences, this study provides new insight for understanding what influences workers' job satisfaction and how it is influenced in the Chinese construction industry, and it enriches the literature on the job satisfaction of construction workers.

## 2. Literature Review

This section explores the multidimensional concept of job satisfaction and analyzes the influencing factors of job satisfaction based on the existing research. Then, this section reviews the relevant papers and identifies seven factors of job characteristics that may affect the job satisfaction of the NGCW. Moreover, considering the personal traits and interest demands of the NGCW, another five factors are determined within the discussion. On this basis, the influencing factors index system of the job satisfaction of the NGCW is established, which lays the foundation for the hybrid modeling process using DEMATEL and ISM.

### 2.1. Job Satisfaction

Job satisfaction comes from employees' comparisons of actual outcomes with those that are desired [27,41,44]. In other words, job satisfaction depends on the degree of fit between job characteristics and employees' expectations [53]. Similarly, job satisfaction is also regarded as a concept based on workers' psychology and perceptions of the external environment [51]. From the perspective of emotional experience, Locke (1976) defined job satisfaction as pleasure or a positive emotional state resulting from a satisfactory appraisal of one's job or job experiences [54]. This emotional state is generated from an individual's overall evaluation of work [55,56], and it has a direct impact on employees' labor market behavior [25,57], organizational citizenship behavior [28,58] and job performance [59,60]. In addition, job satisfaction is defined as psychological responses to the job, and it includes cognitive, emotional and behavioral components [32].

Job satisfaction can be regarded as a global concept, such as Locke's classic definition, which focuses on overall satisfaction [44]. Another view of job satisfaction is a multidimensional concept with personal traits and environmental factors and is composed of facets of satisfaction with various aspects of a job [26]. In addition, the facets of job satisfaction, including pay, colleagues, supervisors, working conditions, job security, promotions and the work itself, are often discussed in the previous literature [44,59,61]. Despite its inconsistent definition, job satisfaction is generally considered a multidimensional concept, including a series of employees' views on work [50]. In line with this argument, job satisfaction can be defined as how the employee feels about his job and various aspects of the job [62]. Moreover, there is considerable empirical evidence that the linear combination of satisfaction with various aspects is a sufficient measure of overall satisfaction [63]. However, Roelen et al. (2008) suggested that the measurement of satisfaction with various aspects should be separated from overall satisfaction [26] because employees are likely to be satisfied with certain aspects of the job and not with others [53], or they may be satisfied with many aspects of the job but overall still feel dissatisfied [26]. Employees can hold varying attitudes about different aspects of the job, which strongly proves the multidimensionality of job satisfaction [62].

A few studies have been carried out to understand what influences workers' job satisfaction, mostly focusing on situational factors (including job characteristics and job conditions), and the role of individual features or personal attributes on job satisfaction

is often neglected [28,64]. The former refers to these factors that are related to workers' feelings about various aspects of the job, such as working environment, salary and benefits, promotion opportunities, relationships with co-workers and supervisors and support from the organization [44,59,61]. Moreover, job satisfaction also depends on workers' expectations according to the above definition, which may be affected by individual differences (e.g., gender, age, educational level, growth background) [51,53]. For example, Guglielmi et al. (2016) argued that the age difference of workers should be taken into account in order to design appropriate human resource practices to promote job satisfaction [65]. Moreover, Green and Tsitsianis (2005) pointed out that generational differences may cause changes in job satisfaction [66]. The NGCW grew up in an important period of China's economic and educational reform and development, and it has typical characteristics and needs, such as better education, a stronger awareness of rights protection, pursuing higher-order needs and paying more attention to social status [5,37,40,42]. In addition, the identity of migrant workers and the urban–rural household registration system are unique problems in the context of Chinese culture, which correlate with the job satisfaction of the NGCW. In other words, the facets of job satisfaction do not only differ across job characteristics, but they also differ across personal traits and cultural backgrounds. Therefore, it is necessary to conduct specific research on the job satisfaction of the NGCW with Chinese cultural backgrounds.

### 2.2. Influencing Factors of Job Satisfaction

It has been confirmed in previous studies that job satisfaction is closely related to personal traits and job characteristics [24] or to job characteristics and workers traits [50,57]. For example, Wang and Jing (2018) thought that the determinants of job satisfaction can be explained by the person effects model and the situation effects model, which correspond with personal traits and job characteristics as well as with social-related factors [67]. Moreover, Tutuncu and Kozak (2008) argued that working conditions, demographic characteristics and expectations of future careers or the type of work can determine the level of job satisfaction [53]. In the construction industry, the significance of the individual attributes of workers and job-related features in shaping job satisfaction and determinants has been recognized in many studies [49]. For example, the influencing factors of construction workers' job satisfaction can be divided into two groups: individual factors and job characteristics [11], or into two main areas: personal determinants and organizational factors [31]. Moreover, job satisfaction is also the worker's attitude or emotional response to the work-related social environment [31]. Roelen et al. (2008) considered job satisfaction as a multidimensional concept with personal traits and environmental factors [26]. It needs to consider job satisfaction from a wider perspective, such as with personal and social factors [48]. In addition, the determinants of job satisfaction are also categorized into work-related and non-work-related factors in the literature [67]. The former includes working environment, job characteristics and work-specific personal factors, and the latter refers to demographic, culture-related and society-related factors. Based on the above views, this study analyzes the influencing factors of job satisfaction with the aspects of job characteristics, personal traits and social environment based on the existing literature.

### 2.2.1. Job Characteristics

Job satisfaction is the result of employees' multidimensional attitudes towards their jobs and working places [53], such as the work itself, supervision, co-workers, opportunities, pay, working conditions and security [31]. In addition, the most important factors of employees' perception of job satisfaction are those related to the nature of jobs [48]. Moreover, Hwang et al. (2019) found that critical features of the job, such as the experience of the job, salary, promotion opportunities, supervision, and co-workers, greatly affect workers' assessment of their work [29]. In addition, job satisfaction is based on the concept of personal evaluation of the job [55]; therefore, when job characteristics meet the needs of employees, they show a higher degree of job satisfaction [59].

Due to the important role of job satisfaction in improving productivity and performance in the work environment [68], some studies on the influencing factors of job satisfaction have been carried out in the construction field, which contribute to understanding how to motivate and maintain employees' job satisfaction. For Indonesian construction workers, job characteristics, rewards and relations with superiors and co-workers are considered to have impacts on their job satisfaction [27]. Moreover, job characteristics are proved to have a significant influence on the job satisfaction of foreign workers in Taiwan's construction industry [69]. On the basis of the theory of motivation, Maslow's hierarchy of needs and two-factor theory, the factors include but are not limited to wage/pay, recognition, supervision, the work itself, security, work environment, co-workers and promotion opportunities, which affect the job satisfaction of workers in the construction industry chain in Ghana [31]. Furthermore, Bakotić (2016) proposed a conceptual model of factors that influence job satisfaction, which includes the nature of work, opportunities for advancement, possibilities for further education, leadership, co-workers, direct supervisors, position in the company, working conditions, permanent employment and work hours [70].

### 2.2.2. Personal Traits

The job characteristics theory proposed by Hackman and Oldham (1980) is an extension of the two-factor theory [71]. It is believed that job satisfaction is affected by internal (motivator/intrinsic) and external (hygiene/extrinsic) factors, as well as by the personal traits of employees. Workers with different personal traits pay different attention to various work aspects and have different expectations, so their job satisfaction is different due to the work aspects that are considered [72]. Although construction workers' expectations are related to personal traits, the relevant studies mainly focus on the productivity of the process rather than on employees' personal psychology [61,73]. As a result, the existing literature often ignores the impact of personal characteristics or attributes on job satisfaction [28,64]. At present, personal characteristics, such as age, education level and gender are considered to affect job satisfaction [31]. For example, Rotimi et al. (2021) explored the influence of demographic factors on the job satisfaction of migrant construction workers in New Zealand [43]. Moreover, Wang and Jing (2018) reviewed the literature about the determinants of job satisfaction of migrant workers, and it was concluded that personal traits include demographics and work-specific personal factors [67].

In addition, previous studies linked traits from the 5-factor model of personality to overall job satisfaction and found that only neuroticism and extraversion are related with job satisfaction, generalized across studies [74]. Based on the two-factor theory, Furnham et al. (2002) further found that neurotics are sensitive to hygiene/extrinsic factors, whereas extraverts are sensitive to motivator/intrinsic factors [75].

### 2.2.3. Social Environment

Although most of the differences in job satisfaction can be explained by job characteristics and personal traits, there are some other factors that deserve researchers' attention. For workers, the job is not only a source of income and an important guarantee for survival, but it is also the way to meet the need for self-actualization, self-esteem and professional respect [42], which is in line with Maslow's need theory. Marzuki et al. (2012) found that meeting high-order needs significantly improves the job satisfaction of Indonesian construction workers [27]. Anin et al. (2015) also suggested that construction managers should take measures in job design to meet the high-order needs of construction workers [31]. Moreover, Bowen and Cattell (2008) found that workplace discrimination based on gender and racial background has a negative impact on the job satisfaction of construction workers [50]. In addition, Wang and Jing (2018) reviewed non-work-related determinants of the job satisfaction of immigrant workers, which include demographic, culture-related and society-related factors [67].

In addition, a job is only a part of life, and professional identity is one of the many roles of workers. However, the project-based industry and unstable employment relationships

make it difficult for construction workers to balance family life, which may result in role conflict. Therefore, the conflict between job and family has always been one of the main concerns in the construction industry. In particular, poor work–family balance also affects the job satisfaction of construction workers. For example, Lingard and Francis (2004) pointed out that work–family conflict caused by long working hours is the main reason for the dissatisfaction of on-site workers, and young workers need to achieve more balance between work and their family [76].

In addition, discrimination and prejudice inside and outside a job may have a negative impact on migrant workers' satisfaction and well-being [67]. Moreover, there are some social problems in the construction industry in China, such as less respectable jobs, the identity of both workers and peasants and the urban–rural dual system, which prevents construction workers from gaining professional respect and work–family balance. As the resolution of these issues involves many social aspects, professional respect and work–family balance are considered environmental factors.

### 2.3. Establishment of the Influencing Factor Index System of Job Satisfaction of the NGCW

The existing literature shows that job characteristics and personal traits can influence workers' job satisfaction, but personality, age, gender, etc., cannot be affected by managers; only job characteristics can be managed [26]. In addition, Spector (1985) believed that personal characteristics are related to job satisfaction, but this relationship is weak and variable [62]. Therefore, the relationship between various aspects of work and job satisfaction is more important.

In view of the contributions of previous studies, this study conducts a scoping review using the review framework proposed by Arksey and O'Malley (2005) [77]. This review aims to understand the influencing factors of the job satisfaction of construction employees. Therefore, this study applies extensive search criteria to extract relevant papers published in peer-reviewed journals. To maximize inclusivity, this study searches five databases, such as Taylor & Francis, Emerald, ASCE, Elsevier and Google Scholar. With discussion within the research team and suggestions from experts, generic search terms are determined to be influencing factors, job-related factors or work factors of job satisfaction. As a result, a total of 4302 papers with abstracts broadly related to the influencing factors of job satisfaction are identified from our search. Moreover, the papers that have a lack of focus on the construction industry are eliminated, and 300 papers remain. Then, this study reviews the remaining 300 papers based on the following exclusion criteria:

- Papers have no explicit focus on the influencing factors of job satisfaction.
- Papers are not published in English.
- Papers are duplicates, or papers are retrieved in two or more databases.
- Papers are less reliable or are of poor quality.
- This study has no access to the full text of the papers.

After finishing the above process, 15 papers are included in the scoping review, which contribute to identifying the factors of job characteristics that may the affect job satisfaction of the NGCW (as shown in Table 1).

**Table 1.** Factors of job characteristics that affect job satisfaction of NGCW.

| Code | Influencing Factors | Description | References |
|------|---------------------|-------------|------------|
| 1 | Career development | Vocational skills training and promotion opportunities for construction workers | [11,25–27,29,49,53,59,70,78,79] |
| 2 | Labor intensity | Working hours and workload of construction workers | [11,25,26,59,70,76,78,79] |
| 3 | Working environment | Working environment of construction workers on site | [26,31,51,59,70,79] |
| 4 | Salaries and rewards | Fixed amount paid to workers for the work performed, which is calculated on a daily, monthly or annual basis | [11,26,27,29,31,49,51,59,70,76,78,79] |
| 5 | Rights protection | Provided labor protection and purchased insurance for construction workers | [25,31,59,70,79] |
| 6 | Colleague relationships | Attitudes of construction workers towards their co-workers | [11,26,27,29,31,51,59,70,78–80] |
| 7 | Leadership style | Supervisor's attitude and behavior towards construction workers | [26,27,29,31,51,53,59,70,78–80] |

However, with the growth of the social economy and the improvement of living standards, the NGCW has better education and higher expectations for work. The relationship between education level and job satisfaction has been confirmed [81]. Specifically, higher education levels may provide better job positions, higher income and more career development opportunities, and the foundation of job satisfaction lies in them [43]. Moreover, Eissenstat et al. (2021) found that education level matching is associated with job satisfaction [82]. Specifically, the mismatch between the education level of workers and their job can reduce job satisfaction [83]. In addition, it is defined as a competency-related factor in a review study by Wang and Jing (2018) [67]. Therefore, decision makers should consider the post competency of workers in job design to ensure that the skills they possess match the job's requirements, which can contribute to adjusting the job satisfaction of workers [84]. In particular, better education may enhance the competency and self-efficacy of workers [43]. Moreover, the NGCW has a strong consciousness of safeguarding rights. In other words, labor rights and other employment-related benefits are more valued than they have previously been [5]. These three personal traits, i.e., education level, competency and the consciousness of safeguarding rights, are not only important characteristics that distinguish the NGCW from the older generation, but they also have a significant influence on job satisfaction. Therefore, this study considers these three personal traits as influencing factors of job satisfaction of NGCWs.

Considering the interest demands and higher-level needs of the NGCW, the role of social environmental factors on their satisfaction cannot be ignored. In other words, these factors contribute to improving job satisfaction of the NGCW through professional respect and work–family balance. This is consistent with the study by Srivastava and Kanpur (2014), which demonstrates that the quality a worker's work–life balance brings them job satisfaction improves their job performance [42]. Moreover, it is common for construction workers to have family separation issues under the pressure of breadwinning [43]. In addition, the situation of social environment factors in China is of great discrepancy to those of other countries. For example, migrant workers face different treatment from urban locals in terms of income, welfare, security, identity, professional respect [85], etc. As a result, this study considers professional respect and work–family balance as social environmental factors of job satisfaction of the NGCW.

In summary, this study identifies 12 main factors that affect the job satisfaction of the NGCW based on a literature review and on the personal traits of the NGCW, namely education level ($S_1$), consciousness of safeguarding rights ($S_2$), competency ($S_3$), career

development ($S_4$), labor intensity ($S_5$), working environment ($S_6$), salaries and rewards ($S_7$), rights protection ($S_8$), colleague relationships ($S_9$), leadership style ($S_{10}$), professional respect ($S_{11}$) and work–family balance ($S_{12}$), as shown in Figure 1.

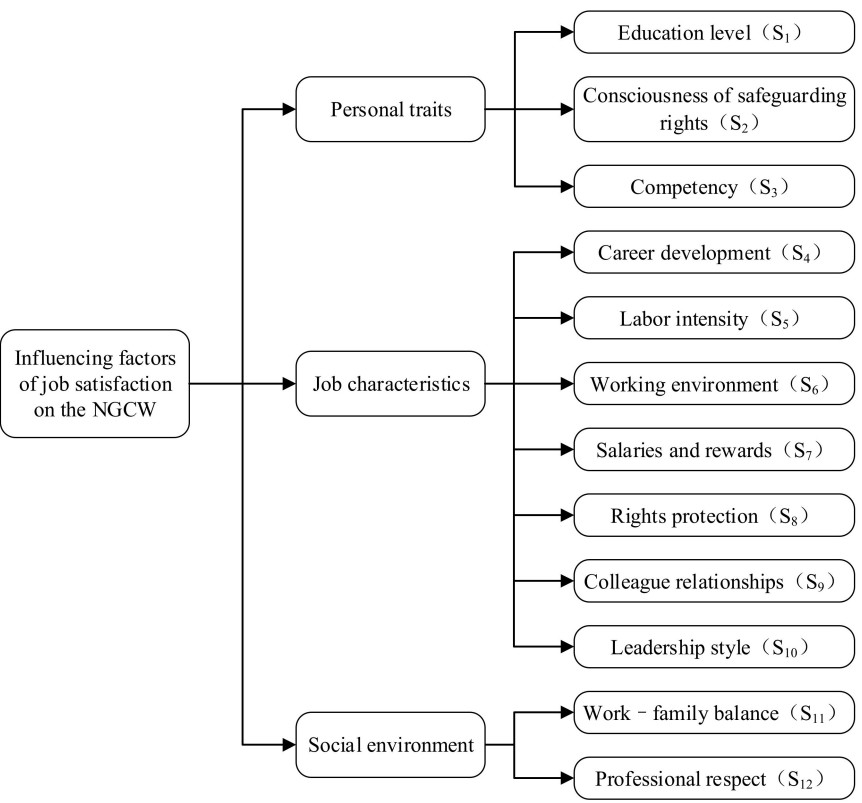

**Figure 1.** Index system of influencing factors of job satisfaction of the NGCW.

## 3. Research Methods

### 3.1. Method and Model

Job satisfaction is influenced by a group of factors that often have interwoven impacts [26]. The interactions among factors are usually non-linear and feed back into each other, which forms multiple interactive loops and extremely complex influence mechanisms, thus making the system uncertain and overall opaque. It is necessary to employ methods in complex scientific field, such as the Decision-Making Trial and Evaluation Laboratory (DEMATEL) and Interpretive Structural Modeling (ISM) [86–90].

DEMATEL and ISM are both system structure modeling methods that use matrix and graph theory, investigating cause and effect relationships among factors on the basis of direct and indirect interactions between any two factors in a system [91,92]. In addition, DEMATEL focuses on analyzing the importance of factors in a system and on categorizing factors into a cause group and an effect group, and the advantage of ISM is establishing a hierarchical structure that can reflect interactions between various factors [91]. Therefore, the ability of DEMATEL to make a complex factor system structured and layered is obviously weaker than ISM. However, a large amount of complicated matrix computations exists when there are many factors in a system while solely applying the ISM method [93].

Therefore, in order to determine the importance and influence mechanisms of the factors, a combination of DEMATEL and ISM is employed in this study to quantitatively calculate the centrality and causality of factors and to establish related interrelationships and hierarchical structure models. On a theoretical basis, specific algorithms were proposed by Zhou and Zhang (2008) [94], and the basic flows of the integration of DEMATEL and ISM are illustrated in Figure 2. Finally, the results obtained by both methods in the study are analyzed to determine the key influencing factors and to verify their reliability.

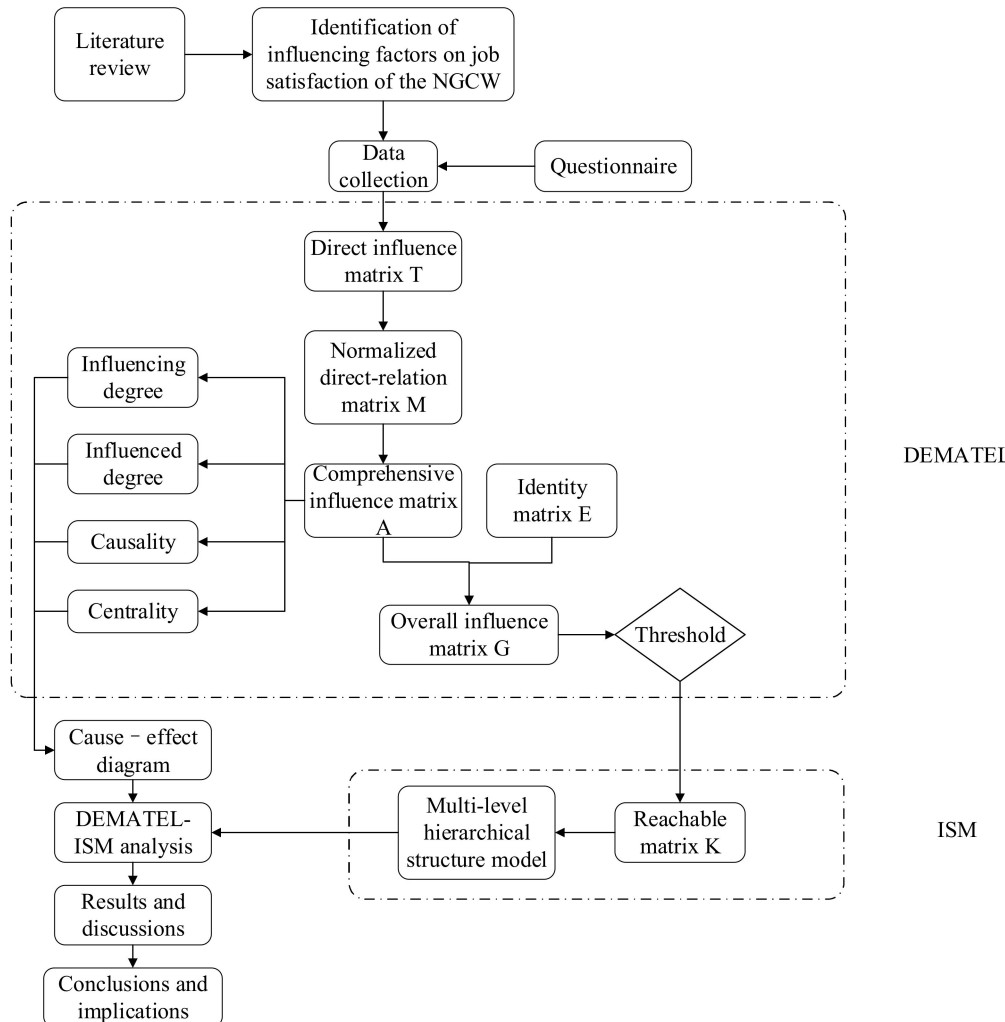

**Figure 2.** Framework for the integration of DEMATAL and ISM.

## 3.2. DEMATEL-ISM Hybrid Modeling Process

### 3.2.1. Data Collection

This paper has identified 12 factors affecting the job satisfaction of the NGCW at three levels, denoted as $S_1$, $S_2$, ... , $S_{12}$. To gather data about the interrelationship and degree of influence among the 12 factors, this paper designs a questionnaire by a focus group discussion with the research team. The questionnaire consists of three sections. The researchers present the objectives of this study and provide the experts with descriptions of the 12 factors to clarify the meaning of each factor in the first section. In addition, the second section captures basic information from survey respondents. Moreover, the influence degree among factors is divided into four grades: no influence (0); low influence (1); high influence (2); and very high influence (3). Considering direct influences between different factors, each expert is required to compare the factors and to score them from 0 to 3 in a paired way to form a pair-wise comparison matrix in the third section.

DEMATEL and ISM depend on the experts' opinions for exploring the relationship among the factors. Therefore, the questionnaires are distributed to a total of 20 experts in engineering management, filed via e-mail and paper form, among which 20 questionnaires are recovered. Before the questionnaire survey, this paper introduces information and background to the participants to ensure they have good knowledge. The expert group includes some professors, associate professors and lecturers of engineering management from the School of Mechanics and Civil Engineering, China University of Mining and Technology, as well as from 10 front-line managers from construction enterprises and

project management companies. The basic information of experts is shown in Table 2. The experts that have at least 10 years of professional experience account for 80%. In other words, they have a solid theoretical foundation and rich practical experience in not only the behavior and mental health of construction workers, but also in engineering management, which may provide this paper the perspectives of experts who deal with the issues about the satisfaction of construction workers in China. Moreover, this paper employs data triangulation to ensure the reliability of the research. Therefore, surveys of each expert are conducted independently to evade any impact one expert may have on another. Then, this paper conducts the DEMATEL-ISM hybrid modeling process based on the data from academics and front-line managers.

**Table 2.** The basic information of experts.

| Variable | Categories | Number of Experts |
|---|---|---|
| Affiliation | University | 10 |
| | Construction enterprises | 4 |
| | Project management companies | 6 |
| Title | Lecturer | 1 |
| | Associate professor | 8 |
| | Professor | 1 |
| | Assistant engineer | 6 |
| | Senior engineer | 3 |
| | Professor level senior engineer | 1 |
| Experience | Less than 5 years | 2 |
| | Between 6 and 10 years | 2 |
| | Between 11 and 15 years | 8 |
| | Between 16 and 20 years | 4 |
| | More than 20 years | 4 |

3.2.2. DEMATEL Analysis

- Establishment of direct influence matrix

All responses from the 20 experts are summarized and calculated according to Equation (1). The direct influence matrix $T = \left[t_{ij}\right]_{n \times n}$ consisting of 12 factors is established and listed in Table 3 (when $i = j$, $t_{ij} = 0$).

$$t_{ij} = \frac{\sum_{k=1}^{m} t_{ij}^k}{m} \tag{1}$$

where $m$ denotes number of experts ($m = 20$) and $t_{ij}^k$ denotes the direct influence of factor $S_i$ on factor $S_j$.

**Table 3.** Direct influence matrix T for influencing factors of job satisfaction.

| Factor | $S_1$ | $S_2$ | $S_3$ | $S_4$ | $S_5$ | $S_6$ | $S_7$ | $S_8$ | $S_9$ | $S_{10}$ | $S_{11}$ | $S_{12}$ |
|---|---|---|---|---|---|---|---|---|---|---|---|---|
| $S_1$ | 0.00 | 2.35 | 2.30 | 2.60 | 1.40 | 1.85 | 2.40 | 1.90 | 1.45 | 1.80 | 2.10 | 1.95 |
| $S_2$ | 1.05 | 0.00 | 1.00 | 1.20 | 1.10 | 1.10 | 1.15 | 2.50 | 1.00 | 1.05 | 1.30 | 1.35 |
| $S_3$ | 1.80 | 1.10 | 2.40 | 0.00 | 1.45 | 1.65 | 2.50 | 1.80 | 1.45 | 1.60 | 1.90 | 1.70 |
| $S_4$ | 1.65 | 1.50 | 1.50 | 1.70 | 1.50 | 1.60 | 2.30 | 1.95 | 1.60 | 1.55 | 2.05 | 2.00 |
| $S_5$ | 1.00 | 1.05 | 1.95 | 2.05 | 0.00 | 1.65 | 2.20 | 1.55 | 0.85 | 1.00 | 1.25 | 1.95 |
| $S_6$ | 1.25 | 1.15 | 1.75 | 1.85 | 1.55 | 0.00 | 1.40 | 1.25 | 1.10 | 1.10 | 1.50 | 1.70 |
| $S_7$ | 1.50 | 1.35 | 1.30 | 1.65 | 1.95 | 1.60 | 0.00 | 1.70 | 1.30 | 1.15 | 1.75 | 2.15 |
| $S_8$ | 1.35 | 2.25 | 1.65 | 2.00 | 1.45 | 1.90 | 1.80 | 0.00 | 1.00 | 1.00 | 1.65 | 1.75 |
| $S_9$ | 0.60 | 1.10 | 1.25 | 1.95 | 0.80 | 1.30 | 1.00 | 1.30 | 0.00 | 1.25 | 1.10 | 1.55 |
| $S_{10}$ | 0.90 | 1.55 | 1.70 | 2.00 | 1.50 | 1.80 | 1.65 | 1.80 | 1.55 | 0.00 | 1.60 | 1.60 |
| $S_{11}$ | 1.10 | 1.65 | 2.30 | 2.60 | 1.20 | 1.75 | 1.40 | 1.80 | 1.25 | 1.40 | 0.00 | 1.50 |
| $S_{12}$ | 1.30 | 1.55 | 1.00 | 1.20 | 1.25 | 1.35 | 1.85 | 1.65 | 1.55 | 1.35 | 1.25 | 0.00 |

- Composition of a normalized direct-relation matrix

The direct influence matrix is calculated according to Equation (2), and the normalized direct-relation matrix $M$ is established. After normalization, the values of elements in matrix $M$ are all between 0 and 1.

$$M = \frac{T}{\max\limits_{1 \leq i \leq n} \sum_{j=1}^{n} t_{ij}} \tag{2}$$

where *max* denotes the maximum value of the sum of all rows in direct influence matrix $T$.

- Establishment of comprehensive influence matrix A

The direct influence matrix can only reflect direct relations among factors, which is far from enough, and indirect relations among factors should be considered [95]. According to Equation (3), the comprehensive influence matrix $A$ is obtained, as shown in Table 4, which can reflect comprehensive relations among factors, including direct and indirect relations.

$$A = M(I - M)^{-1} \tag{3}$$

where $I$ denotes the identity matrix.

**Table 4.** Comprehensive influence matrix A for the influencing factors of job satisfaction.

| Factor | $S_1$ | $S_2$ | $S_3$ | $S_4$ | $S_5$ | $S_6$ | $S_7$ | $S_8$ | $S_9$ | $S_{10}$ | $S_{11}$ | $S_{12}$ |
|---|---|---|---|---|---|---|---|---|---|---|---|---|
| $S_1$ | 0.243 | 0.384 | 0.412 | 0.467 | 0.327 | 0.380 | 0.438 | 0.410 | 0.309 | 0.324 | 0.391 | 0.412 |
| $S_2$ | 0.200 | 0.185 | 0.246 | 0.281 | 0.218 | 0.241 | 0.265 | 0.314 | 0.201 | 0.204 | 0.249 | 0.269 |
| $S_3$ | 0.295 | 0.307 | 0.288 | 0.434 | 0.303 | 0.343 | 0.409 | 0.372 | 0.285 | 0.292 | 0.354 | 0.371 |
| $S_4$ | 0.292 | 0.327 | 0.389 | 0.331 | 0.308 | 0.345 | 0.405 | 0.383 | 0.294 | 0.293 | 0.363 | 0.386 |
| $S_5$ | 0.218 | 0.251 | 0.289 | 0.322 | 0.192 | 0.286 | 0.334 | 0.301 | 0.214 | 0.221 | 0.271 | 0.319 |
| $S_6$ | 0.224 | 0.251 | 0.287 | 0.325 | 0.253 | 0.213 | 0.299 | 0.286 | 0.222 | 0.223 | 0.277 | 0.304 |
| $S_7$ | 0.269 | 0.300 | 0.349 | 0.390 | 0.306 | 0.323 | 0.287 | 0.350 | 0.264 | 0.260 | 0.330 | 0.369 |
| $S_8$ | 0.254 | 0.325 | 0.329 | 0.368 | 0.277 | 0.324 | 0.349 | 0.268 | 0.243 | 0.244 | 0.315 | 0.341 |
| $S_9$ | 0.172 | 0.219 | 0.245 | 0.285 | 0.195 | 0.237 | 0.246 | 0.253 | 0.149 | 0.203 | 0.229 | 0.264 |
| $S_{10}$ | 0.233 | 0.294 | 0.322 | 0.370 | 0.277 | 0.317 | 0.340 | 0.339 | 0.263 | 0.198 | 0.310 | 0.332 |
| $S_{11}$ | 0.227 | 0.283 | 0.289 | 0.348 | 0.250 | 0.298 | 0.311 | 0.321 | 0.237 | 0.244 | 0.225 | 0.309 |
| $S_{12}$ | 0.243 | 0.287 | 0.317 | 0.362 | 0.260 | 0.291 | 0.339 | 0.324 | 0.257 | 0.250 | 0.288 | 0.256 |

- Calculation of the influencing degree, influenced degree, centrality and causality

The influencing degree, denoted as $R_i$, represents the comprehensive influence of factor $S_i$ on other factors, which is equal to the sum of all elements in the corresponding row of matrix $A$. The influenced degree, denoted as $D_i$, represents the comprehensive influence of other factors on factor $S_i$, which is equal to the sum of all elements in the corresponding column of matrix $A$. The sum of $R_i$ and $D_i$ shows the importance of factor $S_i$ in the system and is called centrality, denoted as $m_i$. The difference between $R_i$ and $D_i$ reflects the pure influence of factor $S_i$ on other factors and is called causality, denoted as $n_i$.

As stated above, the influencing degree $R_i$, influenced degree $D_i$, centrality $m_i$ and causality $n_i$ of various factors on job satisfaction of the NGCW are calculated and shown in Table 5.

**Table 5.** Influencing degree, influenced degree, centrality and causality of factors.

| Factors | Influencing Degree $R_i$ | Ranking | Influenced Degree $D_i$ | Ranking | Centrality $m_i$ | Ranking | Causality $n_i$ | Ranking |
|---------|---------|---------|---------|---------|---------|---------|---------|---------|
| $S_1$ | 4.498 | 1 | 2.869 | 12 | 7.366 | 6 | 1.629 | 1 |
| $S_2$ | 2.872 | 11 | 3.415 | 8 | 6.287 | 11 | −0.542 | 12 |
| $S_3$ | 4.054 | 3 | 3.761 | 5 | 7.814 | 3 | 0.293 | 3 |
| $S_4$ | 4.116 | 2 | 4.284 | 1 | 8.399 | 1 | −0.168 | 5 |
| $S_5$ | 3.215 | 9 | 3.164 | 9 | 6.379 | 10 | 0.051 | 4 |
| $S_6$ | 3.162 | 10 | 3.599 | 7 | 6.761 | 8 | −0.436 | 10 |
| $S_7$ | 3.796 | 4 | 4.021 | 2 | 7.817 | 2 | −0.226 | 6 |
| $S_8$ | 3.637 | 5 | 3.920 | 4 | 7.557 | 4 | −0.284 | 9 |
| $S_9$ | 2.698 | 12 | 2.939 | 11 | 5.636 | 12 | −0.241 | 7 |
| $S_{10}$ | 3.594 | 6 | 2.956 | 10 | 6.550 | 9 | 0.638 | 2 |
| $S_{11}$ | 3.344 | 8 | 3.602 | 6 | 6.945 | 7 | −0.258 | 8 |
| $S_{12}$ | 3.473 | 7 | 3.929 | 3 | 7.402 | 5 | −0.457 | 11 |

- Developing a cause–effect diagram

The horizontal axis vector (R+D), named the 'prominence', is denoted as the centrality m, which reveals the importance of the factors. Similarly, the vertical axis (R−D), named the 'relation', is denoted as the causality n, which can divide factors into cause factors ($n > 0$) and effect factors ($n < 0$). Therefore, a cause–effect diagram can be developed by mapping the dataset of ($m_i$, $n_i$), as shown in Figure 3, providing valuable sight for determining the importance and interrelationships of factors [96].

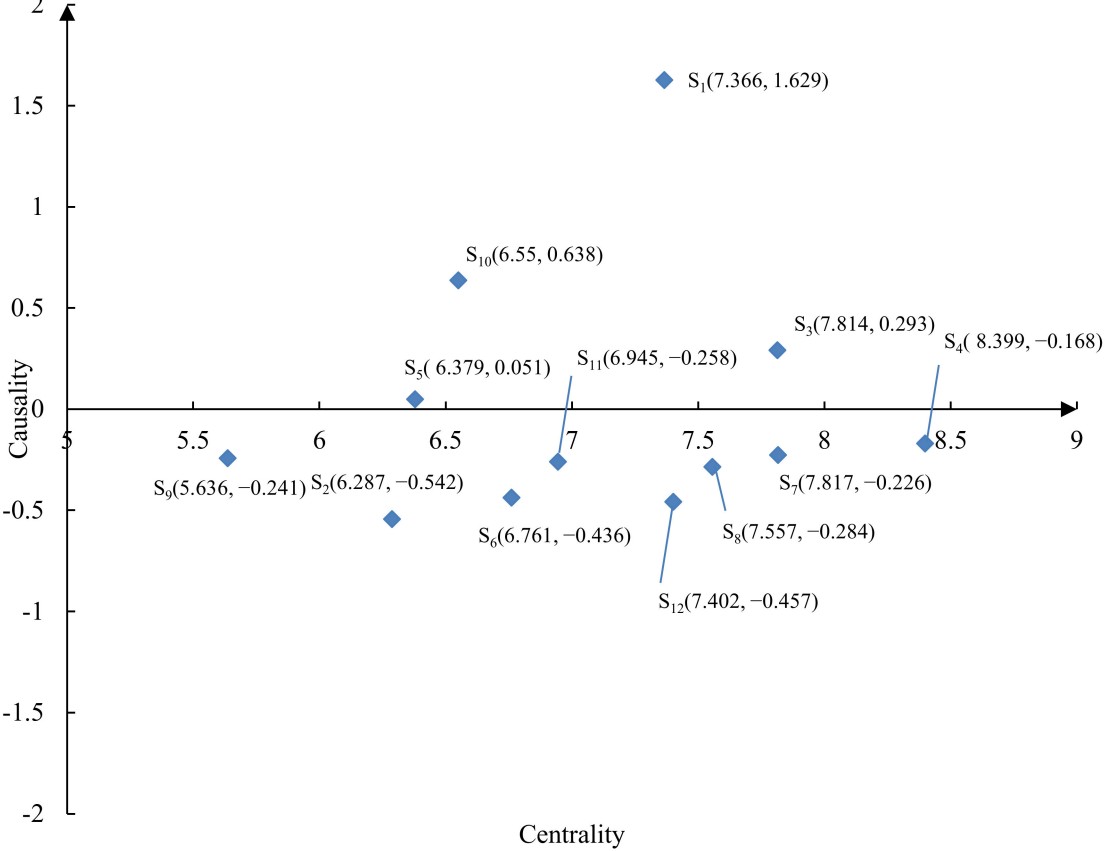

**Figure 3.** Cause–effect diagram.

### 3.2.3. ISM Analysis

- Integrating DEMATEL-ISM to establish a reachability matrix

A comprehensive influence matrix can only reflect the overall interactions among different factors, and it does not consider the influence degree or relation of the individual factor on itself, which can be shown by the identity matrix *I* [93,95]. Therefore, an overall influence matrix *G* is calculated according to Equation (4).

$$G = A + I = [g_{ij}]_{n \times n} \tag{4}$$

The threshold $\lambda$ is introduced to eliminate weak relations among factors and redundant information in the system, which can simplify system structure and obtain a clear hierarchy [97]. However, over-simplification of causality may lead to the neglect of indirect or nonlinear interactions among factors in the system [98]. Therefore, how to determine the threshold $\lambda$ is the key to establishing the reachability matrix.

There are three main methods to determine the threshold $\lambda$ in existing research. The first one is determined by experts and decision makers considering empirical knowledge and practical problems [93,95,99], but it relies too much on subjective judgment. The second one is an averaging method, which determines the threshold $\lambda$ by calculating the average of elements in a comprehensive influence matrix *A* [100–102]. The third one is based on statistical distribution, considering the internal law of normal data distribution, which is an extension of the averaging method [97].

The third one is employed to determine the value of $\lambda$, and a suitable threshold model is established based on many test results, as shown in Equation (5).

$$\lambda = \alpha + \beta \tag{5}$$

where $\alpha$ denotes average value of all elements in comprehensive matrix *A* and $\beta$ denotes the standard deviation of all elements in comprehensive matrix *A*.

In this paper, the average value $\alpha$ is 0.295, and the standard deviation $\beta$ is 0.060. Therefore, $\lambda = \alpha + \beta = 0.355$. Then, the reachability matrix $K = [k_{ij}]_{n \times n}$, consisting of 0s and 1s (as shown in Table 6), is established based on Equations (6) and (7).

$$\text{If } a_{ij} \succ \lambda (i, j = 1, 2, 3, \cdots, n), k_{ij} = 1 \tag{6}$$

$$\text{If } a_{ij} \prec \lambda (i, j = 1, 2, 3, \cdots, n), k_{ij} = 0 \tag{7}$$

where $k_{ij}$ denotes an element in the reachability matrix;
$a_{ij}$ denotes an element in the comprehensive influence matrix;
1 represents a strong relation between two factors;
0 represents no relation or a weak relation between two factors.

**Table 6.** Reachability matrix *K* for the influencing factors of job satisfaction.

| Factor | $S_1$ | $S_2$ | $S_3$ | $S_4$ | $S_5$ | $S_6$ | $S_7$ | $S_8$ | $S_9$ | $S_{10}$ | $S_{11}$ | $S_{12}$ |
|--------|-------|-------|-------|-------|-------|-------|-------|-------|-------|----------|----------|----------|
| $S_1$ | 1 | 1 | 1 | 1 | 0 | 1 | 1 | 1 | 0 | 0 | 1 | 1 |
| $S_2$ | 0 | 1 | 0 | 0 | 0 | 0 | 0 | 0 | 0 | 0 | 0 | 0 |
| $S_3$ | 0 | 0 | 1 | 1 | 0 | 0 | 1 | 1 | 0 | 0 | 0 | 1 |
| $S_4$ | 0 | 0 | 1 | 1 | 0 | 0 | 1 | 1 | 0 | 0 | 1 | 1 |
| $S_5$ | 0 | 0 | 0 | 0 | 1 | 0 | 0 | 0 | 0 | 0 | 0 | 0 |
| $S_6$ | 0 | 0 | 0 | 0 | 0 | 1 | 0 | 0 | 0 | 0 | 0 | 0 |
| $S_7$ | 0 | 0 | 0 | 1 | 0 | 0 | 1 | 0 | 0 | 0 | 0 | 1 |
| $S_8$ | 0 | 0 | 0 | 1 | 0 | 0 | 0 | 1 | 0 | 0 | 0 | 0 |
| $S_9$ | 0 | 0 | 0 | 0 | 0 | 0 | 0 | 0 | 1 | 0 | 0 | 0 |
| $S_{10}$ | 0 | 0 | 0 | 1 | 0 | 0 | 0 | 0 | 0 | 1 | 0 | 0 |
| $S_{11}$ | 0 | 0 | 0 | 0 | 0 | 0 | 0 | 0 | 0 | 0 | 1 | 0 |
| $S_{12}$ | 0 | 0 | 0 | 1 | 0 | 0 | 0 | 0 | 0 | 0 | 0 | 1 |

- Establishment of a multi-level hierarchical structure model

The reachable set and antecedent set of factors should firstly be determined. The reachable set of factor $S_i$, denoted as $R(S_i)$, consists of elements corresponding to columns with a 1 in the *i*-th row of reachability matrix $K$ (as shown in Equation (8)). Similarly, the antecedent set of factor $S_i$, denoted as $P(S_i)$, consists of elements corresponding to rows with a 1 in the i-th column of reachability matrix $K$ (as shown in Equation (9)).

$$R(S_i) = \left\{ S_j / S_j \in S, k_{ij} = 1 \right\}, (i = 1, 2, 3, \cdots, n) \tag{8}$$

$$P(S_i) = \left\{ S_j / S_j \in S, k_{ij} = 1 \right\}, (i = 1, 2, 3, \cdots, n) \tag{9}$$

As noted above, the reachable set $R(S_i)$ and antecedent set $P(S_i)$ of factors can be determined according to Equations (8) and (9). The intersection of both of these sets can be also calculated by using Equation (10). The results are shown in Table 7.

$$C(S_i) = R(S_i) \cap P(S_i), (i = 1, 2, 3, \cdots, n) \tag{10}$$

**Table 7.** Reachable set and antecedent set of factors.

| Factor | Reachable Set $R(S_i)$ | Antecedent Set $P(S_i)$ | Intersection Set $C(S_i)$ | Level |
|---|---|---|---|---|
| $S_1$ | 1, 2, 3, 4, 6, 7, 8, 11, 12 | 1 | 1 | |
| $S_2$ | 2 | 1, 2 | 2 | 1 |
| $S_3$ | 3, 4, 7, 8, 12 | 1, 3, 4, | 3, 4 | |
| $S_4$ | 3, 4, 7, 8, 11, 12 | 1, 3, 4, 7, 8, 10, 12 | 3, 4, 7, 8, 12 | |
| $S_5$ | 5 | 5, | 5 | 1 |
| $S_6$ | 6 | 1, 6 | 6 | 1 |
| $S_7$ | 4, 7, 12 | 1, 3, 4, 7 | 4, 7 | |
| $S_8$ | 4, 8 | 1, 3, 4, 8 | 4, 8 | 1 |
| $S_9$ | 9 | 9 | 9 | 1 |
| $S_{10}$ | 4, 10 | 10 | 10 | |
| $S_{11}$ | 11 | 1, 4, 11 | 11 | 1 |
| $S_{12}$ | 4, 12 | 1, 3, 4, 7, 12 | 4, 12 | 1 |

If Equation (11) is valid, factor $S_i$ can be divided into level 1 and is given the top position in the hierarchical model, which means that other factors can reach factor $S_i$ and that factor $S_i$ cannot reach other factors.

$$R(S_i) = C(S_i), (i = 1, 2, 3, \cdots, n) \tag{11}$$

From Table 7, it is not hard to find that factors $S_2$, $S_5$, $S_6$, $S_8$, $S_9$, $S_{11}$ and $S_{12}$ form the set of factors in level 1, denoted as $L_1 = \{S_2, S_5, S_6, S_8, S_9, S_{11}, S_{12}\}$.

The rows and columns corresponding to the factors in level 1 should be separated out from the reachability matrix because they do not help to achieve nor reach any other factors above their own level [103]. Similarly, iterations continue to repeat the above steps until determining the levels of all factors. Finally, factors are divided into four levels, denoted as $L_1 = \{S_2, S_5, S_6, S_8, S_9, S_{11}, S_{12}\}$, $L_2 = \{S_4, S_7\}$, $L_3 = \{S_3, S_{10}\}$ and $L_4 = \{S_1\}$, as shown in Table 8.

From the final level of the partition, the multi-level hierarchical structure model of influencing factors on the job satisfaction of the NGCW is established, as shown in Figure 4. Interactions among factors are marked by an arrow from factor $S_i$ to factor $S_j$ in the model, which is known as the directed graph or digraph [104–106]. Therefore, it can clearly reflect the hierarchy of various influencing factors and the related influencing mechanisms.

**Table 8.** Final iteration level partition of factors.

| Factor Code | Factors | Level |
|---|---|---|
| $S_1$ | Education Level | 4 |
| $S_2$ | Consciousness of Safeguarding Rights | 1 |
| $S_3$ | Competency | 3 |
| $S_4$ | Career Development | 2 |
| $S_5$ | Labor Intensity | 1 |
| $S_6$ | Working Environment | 1 |
| $S_7$ | Salaries and Rewards | 2 |
| $S_8$ | Rights Protection | 1 |
| $S_9$ | Colleague Relationships | 1 |
| $S_{10}$ | Leadership Style | 3 |
| $S_{11}$ | Professional Respect | 1 |
| $S_{12}$ | Work–Family Balance | 1 |

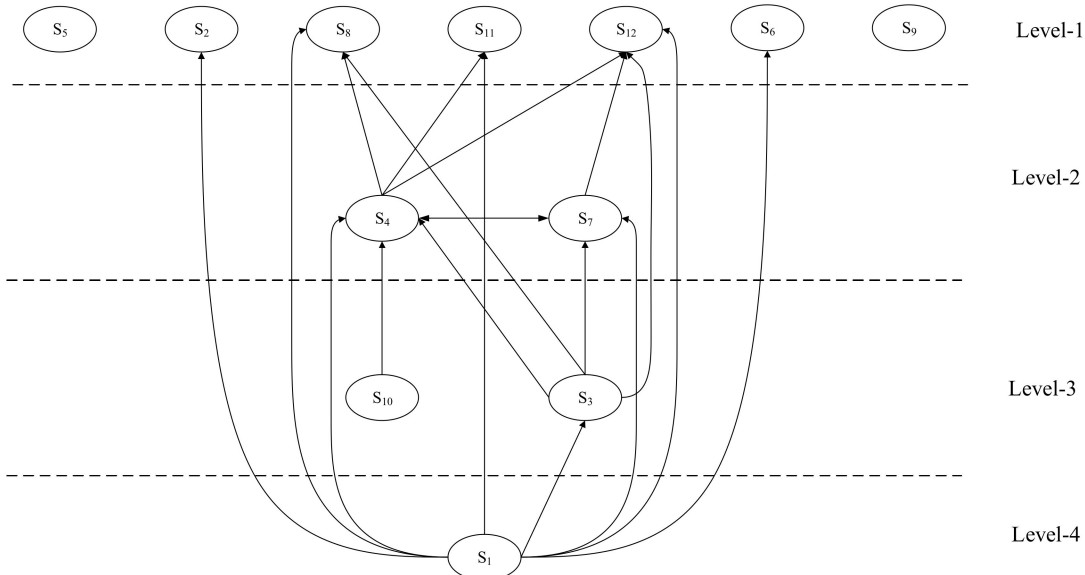

**Figure 4.** Multi-level hierarchical structure model.

## 4. Results of DEMATEL-ISM Hybrid Modeling

### 4.1. Importance and Causal Relations among Various Factors

These factors have quite different influencing degrees, ranging from 2.698 to 4.498, which reflect the comprehensive influence of factor $S_i$ on other factors. As shown in Table 5, education level ($S_1$) is the most influential factor, followed by career development ($S_4$), competency ($S_3$), salaries and rewards ($S_7$), rights protection ($S_8$), leadership style ($S_{10}$), work–family balance ($S_{12}$), professional respect ($S_{11}$), labor intensity ($S_5$), working environment ($S_6$), consciousness of safeguarding rights ($S_2$) and colleague relationships ($S_9$). The influenced degree $D_i$ denotes the comprehensive influence of other factors on factor $S_i$. Eight factors with relatively high influenced degrees include career development ($S_4$), salaries and rewards ($S_7$), work–family balance ($S_{12}$), rights protection ($S_8$), competency ($S_3$), professional respect ($S_{11}$), working environment ($S_6$) and consciousness of safeguarding rights ($S_2$) (as shown in Table 5), which contribute to directly improving the job satisfaction of the NGCW. In particular, competency ($S_3$), career development ($S_4$), salaries and rewards ($S_7$) and rights protection ($S_8$) are believed to be both significant influencers and influences, with high influencing and influenced degrees. Improving these factors can conduce to forming a positive loop for job satisfaction.

Centrality $m_i$ can reflect the total influence/effect of factors on the overall system, representing the position and importance of $S_i$. As shown in the centrality rankings in

Table 5, the priority or relative importance of order of these factors can be given as career development ($S_4$) > salaries and rewards ($S_7$) > competency ($S_3$) > rights protection ($S_8$) > work–family balance ($S_{12}$) > education level ($S_1$) > professional respect ($S_{11}$) > working environment ($S_6$) > leadership style ($S_{10}$) > labor intensity ($S_5$) > consciousness of safeguarding rights ($S_2$) > colleague relationships ($S_9$). Compared with other factors, career development ($S_4$), salaries and rewards ($S_7$) and competency ($S_3$) are ranked first, second and third in regard to the highest centrality value, respectively.

According to the cause–effect diagram (as shown in Figure 3), education level ($S_1$), competency ($S_3$), labor intensity ($S_5$) and leadership style ($S_{10}$) appear under the cause group, with causality rankings of $S_1 > S_{10} > S_3 > S_5$. Education level ($S_1$) is ranked first with the highest causality, most likely because it is an individual characteristic of workers, which is independent from factors of work and working environment. Therefore, education level ($S_1$) is considered to be a crucial factor that has a critical impact on other factors. Likewise, leadership style ($S_{10}$) is related to the personality traits of leaders, which means it is difficult to be influenced by other factors, so its corresponding influenced degree and centrality are ranked lower, ranked at 10th and 9th, respectively (as listed in Table 5). Competency ($S_3$) represents the ability of workers and has significant influence on other factors. In causality, centrality and influencing degree rankings, competency ($S_3$) is the front runner, and its influenced degree is ranked in the middle of all factors. In contrast to the influencing degree of the other factors, labor intensity ($S_5$) is ranked 4th. However, its centrality, causality and influenced degree are all not high, indicating that it is disconnected from other factors in the system.

Career development ($S_4$), salaries and rewards ($S_7$), colleague relationships ($S_9$), professional respect ($S_{11}$), rights protection ($S_8$), working environment ($S_6$), work–family balance ($S_{12}$) and consciousness of safeguarding rights ($S_2$) fall into the effect group with causality values of (−0.168), (−0.226), (−0.241), (−0.258), (−0.284), (−0.436), (−0.457) and (−0.542), respectively. Career development ($S_4$) and salaries and rewards ($S_7$) are front runners for influencing degree, influenced degree, centrality and causality rankings, which means that the two factors have strong linkage with other factors. With low influencing degrees, influenced degrees and centrality, colleague relationships ($S_9$), professional respect ($S_{11}$), working environment ($S_6$) and consciousness of safeguarding rights ($S_2$) play relatively unimportant roles in the system. High centrality of rights protection ($S_8$) suggests that it has a crucial position in the system, but due to both a high influencing degree and influenced degree, the causality is low. A high influenced degree of work–family balance ($S_{12}$) indicates that the factor is vulnerable to other factors and that it shows strong instability. The factor is categorized into the effect group with a low causality ranking. Its outstanding centrality ranking reflects its key role in the system.

In summary, education level, competency, career development, salaries and rewards, rights protection and work–family balance are considered to be critical influencing factors of the job satisfaction of the NGCW.

### 4.2. Interrelationships and Influence Mechanisms among Various Factors

As shown in Figure 4, the multi-level hierarchical structure model of influencing factors of job satisfaction of the NGCW is divided into four layers from bottom to top, including the root layer (level-4), controllable layer (level-3), key layer (level-2) and direct layer (level-1).

- Root layer

Education level ($S_1$) lies in the root layer, which is the starting point of the ISM model and is also the most fundamental but often overlooked factor. It has a strong influence on the job satisfaction of the NGCW, and the influence is continuous. Education level ($S_1$) directly affects competency ($S_3$) in the controllable layer and career development ($S_4$), and salaries and rewards ($S_7$) in the key layer in turn affects factors in the direct layer. Education level ($S_1$) also has a direct influence on consciousness of safeguarding rights ($S_2$), working

environment ($S_6$), rights protection ($S_8$), professional respect ($S_{11}$) and work–family balance ($S_{12}$) in the direct layer.

- Controllable layer

Leadership style ($S_{10}$) directly affects career development ($S_4$) in the key layer, and it affects job satisfaction through the direct layer. Competency ($S_3$) depends on education level ($S_1$) in the root layer and can directly influence career development ($S_4$), salaries and rewards ($S_7$) in the key layer and rights protection ($S_8$), and work–family balance ($S_{12}$) in the direct layer.

- Key layer

Career development ($S_4$) and salaries and rewards ($S_7$) lie in key layer and are at the center of the ISM model, playing the role of connecting factors in the system. In particular, career development ($S_4$) depends on education level ($S_1$) in the root layer and competency ($S_3$) as well as leadership style ($S_{10}$) in the controllable layer. Any action on ($S_4$) also directly affects rights protection ($S_8$), professional respect ($S_{11}$) and work–family balance ($S_{12}$) in the direct layer.

- Direct layer

These factors in the direct layer have weak interrelationships with other factors in the system. For example, labor intensity ($S_5$) and colleague relationship ($S_9$) have no incidence arrow (as shown in Figure 4), which means they are less affected by other factors. This is mainly caused by the set value of threshold $\lambda$ when building the reachability matrix, and weak relations among factors are eliminated. The other five factors are influenced by factors in the deeper layer, indicating that any action on these factors is premised on the resolution of the factors they depend on. In general, these factors have limited connections with the system, but they are the direct cause of job satisfaction and are the most perceivable ones in the analysis.

### 4.3. DEMATEL-ISM Analysis

In DEMATEL analysis, education level $S_1$ falls into the cause group with the highest causality. In ISM analysis, education level ($S_1$) lies in the root layer (the starting point of the hierarchical model) and is considered the root cause of other factors. Similarly, career development ($S_4$) is ranked first with the highest centrality value, and it is an important factor that has close interactions with other factors. It can be seen that there are various arrows of inflow/outflow of $S_4$ in the ISM model. The results obtained from DEMATEL and ISM show the key node location of $S_4$ in the system. Factors under the effect group include career development ($S_4$), salaries and rewards ($S_7$), colleague relationships ($S_9$), professional respect ($S_{11}$), rights protection ($S_8$), working environment ($S_6$), work–family balance ($S_{12}$) and consciousness of safeguarding rights ($S_2$), and they lie in level 1 and level 2 in the ISM model, depending on factors in the deeper level. The cause factors (except labor intensity ($S_5$)) are located at the bottom of the ISM model, which can affect job satisfaction by influencing other factors. Therefore, the results obtained from DEMATEL and ISM methods are consistent to some extent, which means that the framework for the integration of DEMATAL and ISM is reliable and under control.

In summary, the combined DEMATEL and ISM results not only determine the importance and causality of factors but also establish influence mechanisms among various factors and the hierarchical structure model (as shown in Figure 5).

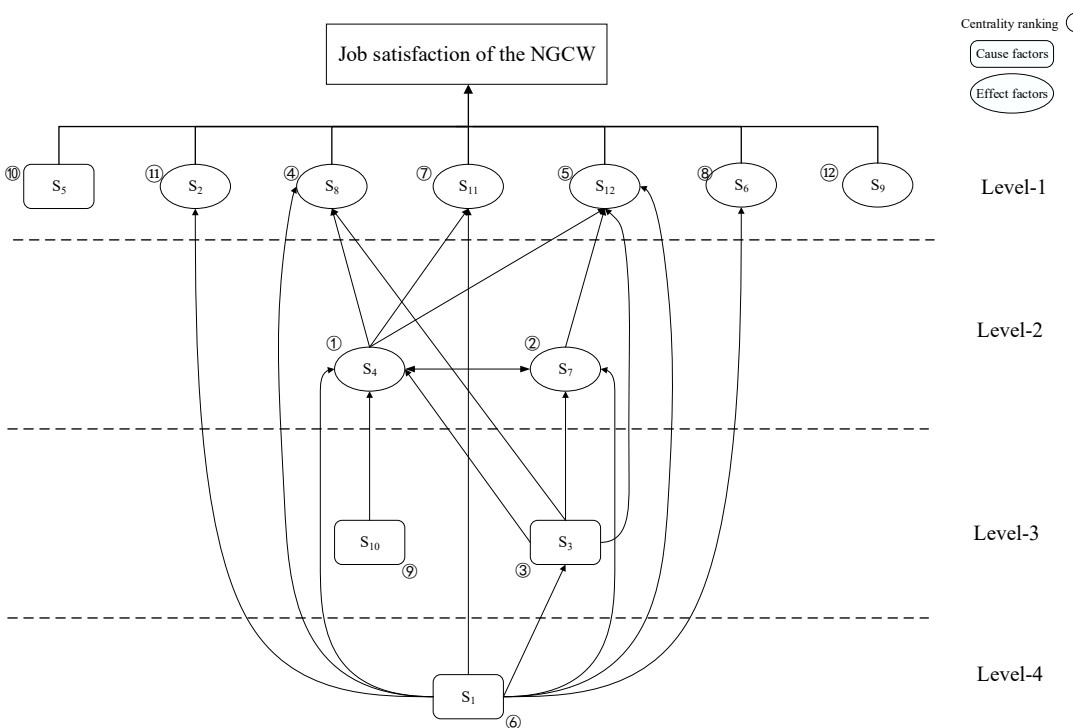

**Figure 5.** Combined DEMATEL-ISM model.

## 5. Discussion and Implications

### 5.1. Discussion

5.1.1. Influencing Path of Factors on Job Satisfaction of NGCWs

With the spread of universal compulsory education in China, the education level of people has been greatly improved in recent years [107]. A survey of migrant workers in 2020 issued by the China National Bureau of Statistics also confirms this result by concluding that the proportion of migrant workers with a college degree or above slightly increased. In other words, better education is one of the important characteristics of the NGCW. As shown in Figure 4, it is not difficult to find that education level is located in the root layer, which is the most influential factor. The factor can directly affect consciousness of safeguarding rights, career development, competency, working environment, salaries and rewards, rights protection, professional respect and work–family balance. It plays an important role in improving the job satisfaction of the NGCW.

High educational level makes the NGCW generally have higher expectations and needs [35]. In other words, it prefers to regard the construction industry as a formal occupation that can change living conditions and realize personal pursuits. Moreover, work has become a means for the NGCW to embrace urban life [108]. In addition, the opportunity for skill training has become increasingly important factors to the NGCW in choosing jobs [5]. In practice, the Chinese government has introduced a series of policies to promote vocational education and to provide skill training opportunities, including on-job and off-job training [7,109,110]. As a result, more and more NGCWs are able to receive vocational skill training and obtain certificates, realizing a transformation from manual labor workers to skilled workers.

High education level makes the NGCW pursue equal rights and have a strong consciousness of safeguarding rights [35]. However, protection of workers' rights in the construction industry is still not ideal, such as arrears of salary [111], lack of labor protection measures [112] and inadequate welfare insurance [113], which seriously threaten the job satisfaction of workers. Considering their greater awareness of rights protection and enhanced welfare [5], the government should formulate relevant laws and policies to protect the fundamental rights of workers. Decision makers in the construction industry must

improve labor protection measures and the insurance system. In addition, the construction of grass-roots trade union organizations needs to be taken seriously, which contributes to protecting the legitimate rights and interests of construction workers. Moreover, it is necessary to take measures to reform aspects of the construction employment system, such as labor contracts, which contribute to establishing stable employment.

The educational level of workers is a reflection of their own abilities, and the NGCW is better educated and skilled than its predecessors. This has an impact on the matching degree of workers' abilities with job requirements, i.e., competency. Strong competency may bring workers better career development opportunities. Moreover, career development is influenced by leadership style, and this is because the supportive relationship between foremen and workers can provide workers with corresponding conditions and opportunities, thus encouraging workers to perform at the highest level in work [31,114]. Good career development usually means a high salary, which is the most important reason for an individual to work and the most powerful motivation for construction workers [115]. High salaries can relieve the burden and pressure of construction workers, improve their quality of life and enable them to live in the city with their families in order to achieve work–family balance. This is consistent with the previous studies that suggest that the NGCW has a stronger tendency to settle permanently in cities [41]. However, due to the identity of migrant workers [116], the household registration system is also an important obstacle for the NGCW and their families to integrate into the city [6]. Therefore, the government should deepen the reform of the household registration system and formulate reasonable policies to ensure that their children can enjoy equal educational opportunities in the workplace and to allow workers to enjoy the fun of family life. Moreover, the government should promote the rental housing market and provide affordable housing for the NGCW.

According to previous studies and Maslow's need theory, the NGCW has higher requirements with regard to the social status of work [5]. In other words, professional respect can bring the NGCW job satisfaction. Relevant studies have also confirmed that the relationship between self-esteem needs and job dissatisfaction of construction workers is the strongest. It is consistent with the view of Bowen and Cattell (2008), whose research found that most workplace characteristics significantly related to job satisfaction can be classified as the "need for self-realization" and the "need for respect" [50]. Thus, it is important to promote the transformation of construction workers to industrial workers or professional workers, from a policy perspective. In addition, poor working environments fail to meet the expectations of the NGCW for jobs, which has a negative impact on job satisfaction. It also has been confirmed that adverse construction working conditions have a negative effect on employees' job production and satisfaction. Moreover, the comparative advantage brought by a relatively high income is disappearing due to the poor working environment of construction [5,7]. Therefore, improving poor working conditions and providing enough labor rights are considered a long-term strategy to attract and retain the NGCW [7], including improving accommodations for construction workers [5].

The NGCW shows lower endurance for work in comparison to the previous generation, but high labor intensity is a significant feature of the construction industry, which is mainly reflected in its working hours and workload. To deal with this problem, the construction industry should transform from using traditional or manual methods to modernization, including lean construction [117] and industrialized construction [118]. In particular, mechanical equipment or construction robots can replace workers to complete high-intensity operations under the background of an aging society [119]. In addition, it is common for construction workers to work overtime because of hurried work. In view of this phenomenon, Clark (1996) proposed shadow wages, which indicate how much of a salary needs to increase for an extra working hour to maintain job satisfaction [24]. The study suggested that decision makers should pay attention to the important role of overtime pay, which helps to offset the negative impacts of overtime on job satisfaction. Moreover, social resources in workplaces play an important role in migrant workers' job satisfaction [85].

Most construction workers stay at the construction site all day, and time spent with fellow workers is even longer than that with their families. Harmonious colleague relationships can create a positive environment, thus bringing pleasure and satisfaction [59]. This viewpoint is supported by Bowen and Cattell (2008), who demonstrated that interactions at work are significantly related to job satisfaction [50].

### 5.1.2. Combination of DEMATEL and ISM

Although the majority of results obtained from the DEMATEL and ISM overlap, this paper has also found and evaluated existing discrepancies. For example, working environment ($S_6$) and professional respect ($S_{11}$) have weak interactions with other factors in the system. However, in the case of DEMATEL, their centrality ranking is not low, which can reflect their important position in the system (as shown in Figure 5). This is because the ISM method indicates whether there is an interrelationship between factors based on the macro approach (0, 1), whereas DEMATEL employs a more complex approach (0, 1, 2, 3) to evaluate the strength of interrelationships between any two factors in the system [120–122]. Moreover, for integrating DEMATEL-ISM to establish a reachability matrix, the threshold $\lambda$ is introduced to simplify the casual relationships among factors in the system, which may eliminate indirect or nonlinear interactions [98].

### 5.2. Research Implications
### 5.2.1. Theoretical Implications

Job satisfaction has a significant impact on labor market behavior [27,29], productivity [51,79], and performance [123,124], which have been discussed by existing studies as the consequences of job satisfaction, and they have made remarkable results. Researching the antecedents of job satisfaction can determine its key influencing factors, which contribute to adopting appropriate and specific human resource practices to attract and retain employees [32,123]. Moreover, previous research findings have shown that the shortage of skilled labor in the construction industry is caused by job attractiveness and job satisfaction [125]. Therefore, researchers gradually follow with the interest in the influencing factors of employees in the construction industry [29,49,51]. This study attempts to explore the influencing factors of job satisfaction from the perspective of the NGCW for the first time, to clarify the interaction mechanisms and hierarchical structures of the influencing factors of the NGCW and to determine critical factors. This is an important theoretical contribution to job satisfaction based on the background of the NGCW in China.

The current studies on the influencing factors of job satisfaction mainly focus on five factors: pay, the work itself, supervision, co-worker relations and promotion opportunities [126]. However, these factors belong to situational factors, including job characteristics and job conditions. There is little literature on the impact of individual features on job satisfaction. According to existing research, the influence of key personal traits such as knowledge, skill and job fit on job satisfaction has drawn attention [29,49]. However, the influencing factors of job satisfaction not only differ across industrial sectors, such as with nurses [127] and bank employees [128], but they also differ from identity, such as with project managers [29], engineers [51] and construction workers [27]. Therefore, by combining generational differences, this study determines factors such as education level, competency and consciousness of safeguarding rights, which are unique to the NGCW in China. Education level and competency are critical factors, which lie in level 1 and level 2, respectively, and they have a continuous influence on the job satisfaction of the NGCW through job characteristic factors.

Considering the changes in needs of the NGCW, this study analyzes unique factors such as professional respect and work–family balance, which complement the social environmental factors of job satisfaction. The construction workers in China are mainly migrant workers, and their status as both workers and farmers makes them not recognized and respected by society. Moreover, the urban–rural dual system in China is an important reason for the work–family conflicts of construction workers. Although professional respect

and work–family balance have been discussed in previous studies, this study considers them to be social environmental factors, which contain contents unique to the NGCW in China.

In summary, this provides a reference for research on job satisfaction, and it is necessary to attach importance to intergenerational differences.

### 5.2.2. Practical Implications

In the context of aging and labor shortage, decision makers in Chinese construction enterprises and the construction industry should consider improving the job satisfaction of construction workers in order to establish a sustainable workforce and to improve the productivity of construction workers. From the perspective of intergenerational differences, this study explores and discusses the influence mechanisms of the job satisfaction of the NGCW, which is instructive to the Chinese construction industry. The research findings show that education level, competency, career development, salaries and rewards, rights protection and work–family balance are critical influencing factors. It has important reference value for decision makers to understand what influences job satisfaction and how it is influenced in order to improve the job satisfaction of the NGCW and to promote the sustainable development of the construction industry in China.

Decision makers should attach great importance to factors in the deeper layer, such as education level, leadership style and competency, which contribute to improving job satisfaction of the NGCW from the root. Specifically, decision makers should promote the professionalization of construction workers to provide training opportunities, to protect their legal rights, to deepen the reform of urban and rural household registration, to increase a sense of belonging and work–family balance, to enhance professional identity and respect and to meet the needs brought by intergenerational differences. In addition, it may achieve better effects for adopting appropriate human resource management practices and for improving the job satisfaction of the NGCW by combining different policies, such as the modernization of construction industry, the reformation of the construction employment system, affordable housing, etc.

### 6. Conclusions and Limitations

#### 6.1. Conclusions

(1) This study identifies 12 influencing factors of the job satisfaction of the NGCW from personal traits, job characteristics and social environment, including education level, consciousness of safeguarding rights, career development, competency, labor intensity, working environment, salaries and rewards, rights protection, colleague relationships, leadership style, professional respect and work–family balance.

(2) Education level, competency, labor intensity, and leadership style come under cause factors of NGCWs' job satisfaction, whereas career development, salaries and rewards, colleague relationship, professional respect, rights protection, working environment, work-family balance, and consciousness of safeguarding rights belong to effect factors of NGCWs' job satisfaction.

(3) Education level is located in the root layer and has a continuous impact on the job satisfaction of the NGCW. Leadership style and competency lie in the controllable layer, which are deep influencing factors. Factors in the key layer include career development and salaries and rewards, which have close interrelationships with other factors. Consciousness of safeguarding rights, labor intensity, working environment, rights protection, colleague relationships, professional respect and work–family balance are direct influencing factors.

(4) Education level, competency, career development, salaries and rewards, rights protection and work–family balance are critical influencing factors of the job satisfaction of the NGCW. Decision makers should consider and address these factors on a high priority basis to improve the job satisfaction of the NGCW.

(5)     The results obtained by DEMATEL and ISM are consistent in most cases. This means that the framework for the integration of DEMATAL and ISM is reliable and under control. It can provide a simple and reliable analysis model that plays a vital role in understanding the interaction mechanisms and hierarchical structures of factors.

### 6.2. Limitations and Future Research

The method of DEMATEL-ISM has its own limitations, and the model greatly depends on the subjective judgment and empirical knowledge of experts. Moreover, different stakeholders and experts analyze complex issues from various perspectives. As a result, this may introduce some elements of bias that can affect the final results. In addition, this study mainly qualitatively explores interactions among factors but fails to verify these interrelationships both in size and direction. Therefore, future research can validate the model statistically by using structural equation modeling.

**Author Contributions:** Conceptualization, G.N.; methodology, H.L. and Z.Z.; software, H.L.; validation, G.N. and T.J.; formal analysis, G.N. and H.L.; investigation, H.L., T.J. and H.H.; writing—original draft preparation, H.L.; writing—review and editing, G.N., T.J. and Z.Z.; supervision, G.N. and H.H.; project administration, G.N.; funding acquisition, G.N. All authors have read and agreed to the published version of the manuscript.

**Funding:** This research was funded by the National Natural Science Foundation of China, grant number 72071201 and the Fundamental Research Funds for the Central Universities in China, grant number 2020ZDPYMS30.

**Institutional Review Board Statement:** Not applicable.

**Informed Consent Statement:** Not applicable.

**Data Availability Statement:** All data generated or used in the study can be obtained from the corresponding author upon reasonable request.

**Acknowledgments:** The authors would like to thank all the respondents of the questionnaire survey in this research. They would also like to thank anonymous reviewers for their helpful comments.

**Conflicts of Interest:** The authors declare no conflict of interest.

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
