# Peer review of "Analysis of Factors Influencing the Job Satisfaction of New Generation of Construction Workers in China: A Study Based on DEMATEL and ISM"

_buildings, doi:10.3390/buildings12050609_

Round 1

Reviewer 2 Report

Dear Authors,

I really enjoyed reviewing you valuable article.

I'd prefer to address my comments for better presentation.

Congratulation.

Kindest regards,

Reviewer.

Reviewer 3 Report

The paper addresses a very relevant and emerging issue, in a proper and detailed manner.

However, I do suggest to the Authors to add some methodological explanations to section '2.LITERATURE REVIEW'. In fact, they limit to report the main results, without explaining how they conducted the review. 

First of all, is it a narrative, a scoping or a systematic review?

Moreover:

  • which databases/sources where used to search for the articles of interest?
  • which keywords were used to perform the search? 
  • then, after they retreived a certain number of papers, how did they select the ones that deserved to be included in the review?

I do think that the above-mentioned details are essential to make the literature review reproducible and, thus, scientifically appropriate.

Reviewer 4 Report

Therefore, it is very important to study and explore the influencing factors of NGCWs’ job satisfaction in China's construction industry. This paper aims to determine the influencing factors of job satisfaction of NGCWs through literature research. In general, the paper is clear and well presented. Some specific suggestions that could help you to improve the paper:

  • Lines 39, 44, 67: transform the text in brackets as a reference
  • Line 108: an introduction between section 2 and section 2.1 could help the reader to understand better the content of section 2
  • Lines 142-143: add more info also about factors
  • Lines 144, 226: why you dived in two separate sections [Influencing Factors of Job Satisfaction and Influencing Factors of Job Satisfaction fof NGCWs] ? in section 2.2 you already ad info also about NGCWs- Please consider to review the structure of this part of the paper.
  • Lines 157-159: please explain why you decide to explain the influencing factors in this way
  • Line 274: add references about DEMATEL and ISM

Round 2

Reviewer 3 Report

The Authors have properly addressed the issues previously arosed and the paper has been sufficiently improved.

Author Response

Thanks for your comment. We appreciate your approval of this manuscript and hope this study can make theoretical and practical contributions to the job satisfaction of NGCWs in China.